# An Update of Amphipoda Checklist for the English Channel

**Jean-Claude Dauvin** 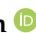

Laboratoire Morphodynamique Continentale et Côtière, Normandie Université, UNICAEN, CNRS, UMR 6143 M2C, 24 Rue des Tilleuls, 14000 Caen, France; jean-claude.dauvin@unicaen.fr

**Abstract:** An updated checklist for 2022 of amphipods from the English Channel (La Manche) is given for eight geographical zones. This revision brings the inventory of 1999 up to date with recent data from the Normano-Breton Gulf and other studies focused on non-indigenous fauna, as well as recent data from benthic and suprabenthic studies, mainly in the eastern part of the English Channel (EC). The total number of species in the entire EC is 269. The number of species is much higher in the western basin (WEC) than it is in the eastern basin (EEC) or in the central part of the EC. The amphipod species listed here are distributed between the eight zones as follows: French WEC: 201 species; English WEC: 194; Normano-Breton Gulf: 224; Bay of Seine: 172; Wight: 97; French EEC: 149; English EEC: 64; and Central EC: 61. Of these species, 180 are present in both basins of the EC, while 78 are present only in the western basin and 11 are present only in the eastern basin. The low number of amphipods (<100) recorded in three of the eight zones is probably due to the lack of observations in these parts of the EC. Among the 269 amphipod species recorded with confidence in the EC, 24 are new to the EC since 1999, 12 are non-indigenous species, and 44 are observed only in one of the eight zones, mainly in the three zones of the western basin of the EC.

**Keywords:** inventory; crustacean; distribution; diversity; non-indigenous species

## 1. Introduction

More than 20 years ago, a checklist of amphipods recorded in the English Channel (EC) was published by [1]. It was based on data from [2,3], inventories of amphipods drawn up by the marine stations of Roscoff [4], Plymouth [5], and Wimereux [6], as well as numerous articles focused on amphipods or the marine fauna in several parts of the EC [7–13]. Moreover, Ref. [14] gives a fully updated list of marine invertebrates, including amphipods, from the Normano-Breton Gulf. Newly developed methods such as experimental plate immersion or scraping of dikes and pontoons have been used to study the sessile and motile species that colonize hard substrates in coastal environments, mainly in Normandy harbours and marinas [9,12]. This has led to a significant increase in our knowledge of these particular environments and has allowed us to detect the presence of several non-indigenous species (NIS) of amphipods. Suprabenthic sledge sampling near the sea bottom in the EC has also facilitated the collection of mobile fauna [7,8].

Moreover, diverse human activities in the EC have led to impacts on the macrobenthic communities. As a consequence, several surveys have been carried out in the Bay of Seine and the eastern part of the EC related to granulate extraction, the dredging and deposition of spoil, and the future implementation of offshore wind farms [15].

Over the past two decades, additional data have become available to revisit and redefine the list of amphipods recorded in the EC. In the present study, amphipod species checklists are given for eight geographical zones of the EC. Previously, Dauvin (1999) [1] provided amphipod checklists for only five zones, where available data were more or less complete due to the existence of Marine Station Inventories, i.e., Roscoff for the western EC along the French coast, Plymouth for the western EC along the British coast, Dinard for the Normano-Breton Gulf, Luc-sur-Mer for the Bay of Seine, and Wimereux for the eastern part

of the EC along the French coast. Moreover, the species checklist presented here includes only those species recorded with confidence in the EC.

## 2. Materials and Methods

The present checklist given for the eight zones (Figure 1) corresponds to the re-arrangement of limits given by [3] for the Fauna of the amphipods around the British Isles, the division of the EC into main zones [15], and the limits proposed by [14] for the Normano-Breton Gulf, which includes the North Cotentin coast as far as the eastern part of the Bay of Cherbourg (Figure 1).

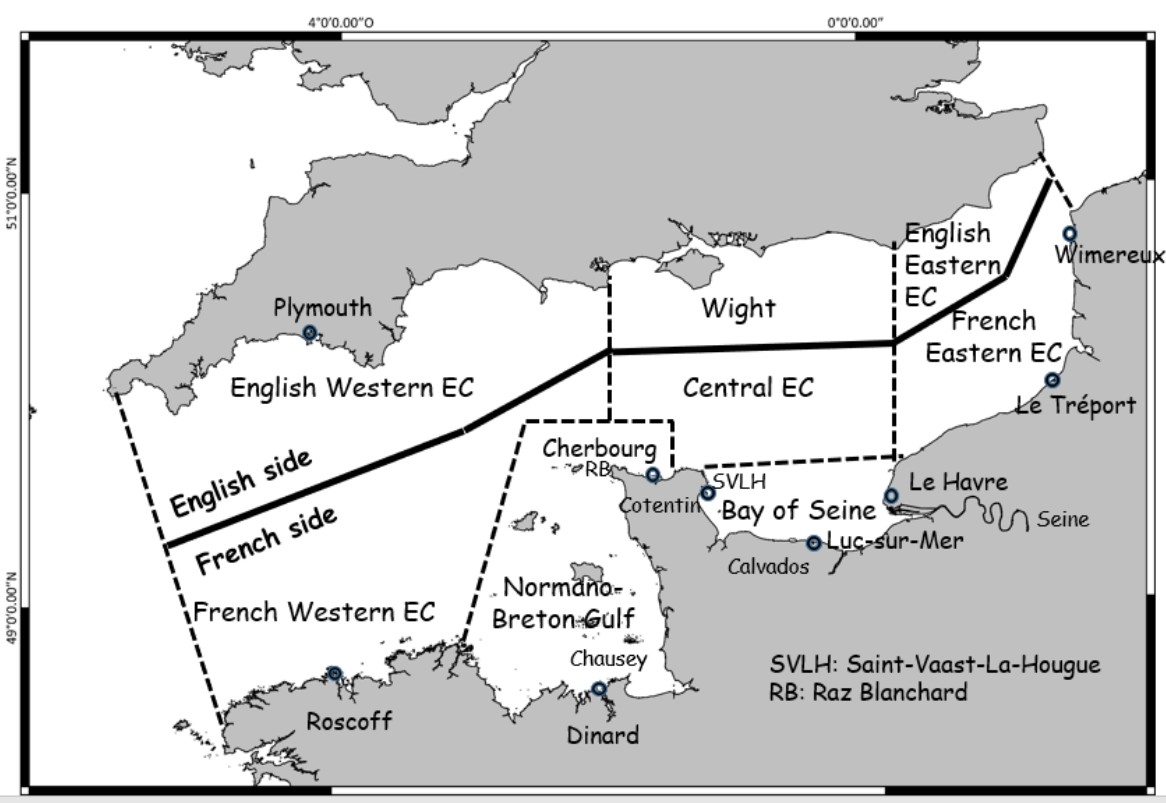

**Figure 1.** Map of the English Channel with the limits of the eight zones and the locations of the main amphipod sampling zones.

(1)　FWEC: French side of the Western English Channel [1,4,16]
(2)　EWEC: English side of the Western English Channel [1,3,5,17,18]
(3)　NBG: Normano-Breton Gulf [1,9,11,13,14,19,20]
(4)　BS: Bay of Seine [1,12] and MABES dataset (https://www.seine-aval.fr/publication (accessed on 1 June 2022))
(5)　CEC: Central English Channel [8,21,22]
(6)　Wight: [3,17,23]
(7)　EEEC: English side of the Eastern English Channel [3,18,24,25]
(8)　FEEC: French side of the Eastern English Channel [1,7,10,24–28]

All the species were checked against the WORMS database of amphipods on 31 May 2022 (https://www.marinespecies.org (accessed on 3 June 2022)). The families and species are ordered alphabetically within the families, and the species are ordered alphabetically with the genera.

To facilitate amphipod identification, Table 1 provides a guide to the fauna as well as publications that can be used to identify the species reported in the checklist of amphipods currently present in the EC. Moreover, it also cites relevant publications on probable species

that are found in neighbouring zones in different sediment types and at the different depths occurring in the English Channel, but which are not yet recorded in the EC.

**Table 1.** Guide of fauna and publications used to identify amphipods recorded in the English Channel and other probable species (found in neighbouring zones in sediment types and depths occurring in the English Channel). 1: [2]; 2: [3]; 3: [29]; 4: [30] and 5: [31]. Column Genus: list of references useful to identify the genus; column Species: list of references useful to identify the species. * Not recorded for the moment in the English Channel.

| Family | General | Genus | Species |
|---|---|---|---|
| **Acidostomatidae** | [2,3] | | |
| **Ampeliscidae** | [2,3,30] | [32] | |
| *Ampelisca aequicornis* Bruzelius, 1859 * | [3,30] | | [33] |
| *Ampelisca armoricana* Bellan-Santini & Dauvin, 1981 | | | [34,35] |
| *Ampelisca cavicoxa* Reid, 1951 * | | | [36] |
| *Ampelisca eclimensis* King, Myers & McGrath, 2004 * | | | [33] |
| *Ampelisca dalmatina* Karaman, 1975 * | | | [30,37] |
| *Ampelisca pectenata* Reid, 1951 | | | [36,38] |
| *Ampelisca provincialis* Bellan-Santini & Kaim-Malka, 1977 * | | | [30,37] |
| *Ampelisca toulemonti* Dauvin & Bellan-Santini, 1982 | | | [38] |
| *Ampelisca sorbei* Dauvin & Bellan-Santini, 1996 * | | | [33,39] |
| *Ampelisca spinifer* Reid, 1951 * | | | [3,30] |
| **Amphilochidae** | [2,3,31] | | |
| **Ampithoidae** | [2,3,31] | | |
| *Ampithoe valida* S.I. Smith, 1873 | | | [40] |
| *Amphitholina cuniculus* (Stebbing, 1874) | | | [41] |
| **Aoridae** | [2,3,31] | | |
| *Aora spinicornis* (Afonso, 1976) * | | | [42] |
| *Aoroides curvipes* Ariyama, 2004 * | | | [43] |
| *Aoroides longimerus* Ren & Zheng, 1996 | | | [43] |
| *Aoroides semicurvatus* Ariyama, 2004 | | | [43] |
| *Grandidierella japonica* Stephensen, 1938 | | | [44] |
| *Lembos denticarpus* Myers and McGratth, 1978 * | | | [45] |
| **Argissidae** | [2,3,31] | | |
| **Aristiidae** | [2,3,31] | | |
| **Atylidae** | [2,3,30] | | |
| **Bathyporeidae** | [2,3,31] | | |
| **Calliopiidae** | [2,3,46] | | |
| **Caprellidae** | [2,46] | | |
| *Caprella mutica* Schurin, 1935 | | | [47,48] |
| *Caprella penantis* (Leach, 1814) | | | [49,50] |
| *Caprella septentrionalis* Krøyer, 1838 | | | [49] |
| *Caprella scaura* Templeton, 1836 * | | | [51] |
| **Cheirocratidae** | [2,3,30] | [52] | |
| *Cheirocratus pseudosundevallii* Gouillieux, 2019 * | | | [52] |

**Table 1.** *Cont.*

| Family | General | Genus | Species |
|---|---|---|---|
| *Cheirocratus robustus* Sars, 1894 * | | | [53] |
| **Cheluridae** | [2,3,30] | | |
| **Colomastigidae** | [2,3,30] | | |
| **Corophiidae** | [2,3,30] | | |
| *Chelicorophium curvispinum* (G.O. Sars, 1895) | | | [54] |
| *Monocorophium uenoi* (Stephensen, 1932)* | | | [55] |
| **Cressidae** | [2,3] | | |
| **Cyproideidae** | [2,3] | | |
| **Dexaminidae** | [2,3,30] | | |
| **Dulichiidae** | [2,3,46] | | |
| **Epimeriidae** | [2,3,30] | [56] | |
| *Epimeria frankei* Beermann & Raupach, 2018 * | | | [56] |
| **Eriopisidae** | [2] | | |
| **Eusiridae** | [2,3,30] | | |
| **Gammarellidae** | [2,3,30] | | |
| **Gammaridae** | [2,3,30] | | |
| *Dikerogammarus villosus* (Sowinsky, 1894) | | | [57] |
| *Gammarus tigrinus* Sexton, 1939 * | [3] | | [3] |
| **Haustoriidae** | [2,3,31] | | |
| **Hyalidae** | [2,3,46] | | |
| *Ptilohyale littoralis* (Stimpson, 1853) | | | [10,58,59] |
| **Hyperiidae** | | [60] | |
| *Hyperia medusarum* (Müller, 1776) | | | [61] |
| *Themisto abyssorum* (Boeck, 1871) | | | [62] |
| *Themisto gaudichaudi* Guérin, 1825 | | | [60] |
| **Iphimediidae** | [2,3] | [63] | |
| *Iphimedia nexa* Myers & McGrath, in Myers, McGrath & Costello, 1987 | | | [63] |
| *Iphimedia perplexa* Myers & McGrath, in Myers, McGrath & Costello, 1987 | | | [63] |
| *Iphimedia spatula* Myers & McGrath, in Myers, McGrath & Costello, 1987 | | | [63] |
| **Isaeidae** | [2,3,31] | | |
| **Ischyroceridae** | [2,3,30] | [64,65] | |
| *Centraloecetes striatus* (Myers & McGrath, 1979) | | | [64] |
| *Ericthonius didymus* Krapp-Schickel, 2013 * | | | [65] |
| *Ericthonius punctatus* (Bate, 1857) | | | [66] |
| *Jassa slatteryi* Conlan, 1990 | | | [40,67] |
| **Lafystiidae** | [2,3,31] | | |
| **Leucothoidae** | [2,3,31] | [68,69] | |
| *Leucothoe denticulata* Costa, 1851 * | | | [70] |

**Table 1.** *Cont.*

| Family | General | Genus | Species |
|---|---|---|---|
| **Liljeborgiidae** | [2,3,31] | [71] | |
| *Idunella dentipalma* (Dauvin & Gentil, 1983) | | | [72] |
| *Idunella mollis* Myers & McGrath, 1983 * | | | [71] |
| *Idunella spinifera* (Dauvin & Gentil, 1983) | | | [72] |
| **Lysianassidae** | [2,3,31] | | |
| **Maeridae** | [2,3,30] | [73] | |
| *Elasmopus thalyae* Gouillieux & Sorbe, 2015 | | | [73] |
| **Megaluropidae** | [2,3,30] | | |
| **Melitidae** | [2,3,30] | [74] | |
| *Melita nitida* Smith, 1873 * | | | [74] |
| **Melphidippidae** | [2,3,46] | | |
| **Microprotopidae** | [2,3,30] | | |
| **Nuuanuidae** | [2,3,30] | | |
| **Odiidae** | [3] | | |
| **Oedicerotidae** | [2,3,46] | [75] | |
| *Pontocrates articus* G.O. Sars, 1895 * | | | [76] |
| *Pontocrates moorei* Myers & Ashelby, 2022 * | | | [75] |
| **Opisidae** | [2,3,31] | [77] | |
| *Normanion chevreuxi* Diviacco & Vader, 1988 | | | [31,77] |
| **Phliantidae** | [2,3,46] | | |
| **Photidae** | [2,3,31] | [78] | |
| *Photis pollex* Walker, 1895 | | | [79] |
| *Photis inornatus* Myers, Rigolet, Thiébaut, Dubois, 2012 * | | | [80] |
| **Phoxocephalidae** | [2,3,46] | [29] | |
| **Pleustidae** | [2,3,46] | | |
| **Podoceridae** | [2,3,46] | | |
| **Scopelocheiridae** | [2,3,31] | | |
| **Stegocephalidae** | [3,46] | | |
| **Stenothoidae** | [2,3,46] | | |
| *Stenothoe eduardi* Krapp-Schickel, 1975 | | | [46] |
| **Synopiidae** | [2,3] | | |
| **Talitridae** | [2,3,46] | | |
| **Tryphosidae** | [2,3,31] | | |
| *Tryphosa crenata* (Chevreux & Fage, 1925) | | | [81] |
| *Tryphosella lowry* Kilgallen, Myers, McGrath, 2006 * | | | [82] |
| **Unciolidae** | [2,3] | | |
| *Uncinotarsus pellucidus* L'Hardy and Truchot, 1964 | | | [83] |
| **Uristidae** | [2,3,31] | | |
| **Urothoidae** | [2,3,31] | | |

Hierarchical Cluster Analysis (HCA) was carried out on the eight zones listed recorded in the EC based on Sorensen's coefficient for the presence/absence of the species found in eight west-east sectors of the EC, with the construction of dendrograms using the group average algorithm generated from the PRIMER V6 software [28].

## 3. Results

### 3.1. Comparison between 1999 and 2022 Checklists

This new 2022 checklist reports 269 amphipod species, i.e., 22 additional species since the last checklist published in 1999 (Table 2).

**Table 2.** Checklist of amphipods in the English Channel. NIS: Non-Indigenous Species. FWEC: French Western English Channel; EWEC: English Western English Channel; NBG: Normano-Breton Gulf; BS: Bay of Seine; CEC: Central English Channel; WIGHT: Wight zone; EEEC: French side of Eastern English Channel; EEEC: English side of Eastern English Channel (see the Figure 1) for the limits of the eight zones of the EC. +: species present in 1999 checklist. *: new location of a species since 1999 for the five zones considered in the first inventory. **: new species for the EC since 1999 checklist.

| | FWEC | EWEC | NBG | BS | CEC | WIGHT | FEEC | EEEC |
|---|---|---|---|---|---|---|---|---|
| **Acidostomatidae** | | | | | | | | |
| *Acidostoma obesum* (Spence Bate, 1862) | + | + | + | + | | | + | + |
| **Ampeliscidae** | | | | | | | | |
| *Ampelisca armoricana* Bellan-Santini & Dauvin, 1981 | + | | * | | | | | |
| *Ampelisca brevicornis* (Costa, 1853) | + | + | + | + | | + | + | + |
| *Ampelisca diadema* (Costa, 1853) | + | + | + | + | | + | + | |
| *Ampelisca pectenata* Reid, 1951 | + | + | + | + | | | + | + |
| *Ampelisca sarsi* Chevreux, 1888 | + | | | | | | | |
| *Ampelisca spinimana* Chevreux, 1900 | + | + | * | | | | | |
| *Ampelisca spinipes* Boeck, 1861 | + | + | + | + | + | + | + | + |
| *Ampelisca tenuicornis* Liljeborg, 1855 | + | + | + | + | | | + | |
| *Ampelisca toulemonti* Dauvin & Bellan-Santini, 1982 | | | ** | | | | | |
| *Ampelisca typica* (Bate, 1853) | + | + | + | + | | | + | |
| **Amphilochidae** | | | | | | | | |
| *Amphilochus manudens* Bate, 1862 | + | + | + | * | + | + | + | + |
| *Amphilochus neapolitanus* (Della Valle, 1893) | + | + | + | + | + | + | + | + |
| *Amphilochus spencebatei* (Stebbing, 1876) | + | + | + | * | | + | | |
| *Amphilochoides boeckii* G.O. Sars, 1895 | | ** | | | | | | |
| *Amphilochoides serratipes* (Norman, 1869) | | ** | | | | | | |
| *Gitana sarsi* Boeck, 1871 | + | + | + | + | + | + | + | |
| *Gitanopsis bispinosa* (Boeck, 1871) | | + | | | | | | |
| *Paramphilochoides intermedius* Scott, 1896) | | ** | | | | | | |
| *Paramphilochoides odontonyx* (Boeck, 1871) | | + | | | | | | |
| **Ampithoidae** | | | | | | | | |
| *Amphitholina cuniculus* (Stebbing, 1874) | + | + | * | | | + | | |
| *Ampithoe gammaroides* (Bate, 1856) | + | + | + | + | | | | |
| *Ampithoe ramondi* Audouin, 1826 | + | + | + | + | | + | | |
| *Ampithoe rubricata* (Montagu, 1818) | + | + | + | + | | + | + | |
| *Ampithoe valida* S.I. Smith, 1873—NIS | | | | ** | | | | |
| *Pleonexes helleri* (Karaman, 1975) | | + | + | * | | + | * | |
| *Suamphitoe pelagica* (Milne Edwards, 1830) | + | + | + | + | | + | + | |
| **Aoridae** | | | | | | | | |
| *Aora gracilis* (Bate, 1857) | + | + | + | + | + | + | + | |
| *Aora spinicornis* (Afonso, 1976) | ** | | | | | | | |
| *Aoroides longimerus* Ren & Zheng, 1996—NIS | | | ** | ** | | | | |
| *Aoroides semicurvatus* Ariyama, 2004—NIS | | | ** | ** | | | | |
| *Grandidierella japonica* Stephensen 1938—NIS | | | | ** | | ** | | |
| *Lembos websteri* Bate, 1857 | + | + | + | | | + | | |

**Table 2.** *Cont.*

| | FWEC | EWEC | NBG | BS | CEC | WIGHT | FEEC | EEEC |
|---|---|---|---|---|---|---|---|---|
| *Microdeutopus anomalus* (Rathke, 1843) | + | + | + | + | | + | | |
| *Microdeutopus chelifer* (Bate, 1862) | + | + | + | + | | + | | |
| *Microdeutopus gryllotalpa* Costa, 1853 | | + | + | + | | + | | |
| *Microdeutopus stationis* (Della Valle, 1893) | + | | + | | | | | |
| *Microdeutopus versiculatus* (Bate, 1857) | + | + | + | + | | | | |
| **Argissidae** | | | | | | | | |
| *Argissa hamatipes* (Norman, 1869) | + | + | * | + | | + | + | |
| **Aristiidae** | | | | | | | | |
| *Perrierella audouiniana* (Spence Bate, 1857) | + | + | + | + | + | | + | + |
| **Atylidae** | | | | | | | | |
| *Atylus falcatus* (Metzger, 1871) | + | + | + | + | | | + | |
| *Atylus guttatus* Costa, 1853 | + | + | + | + | | + | + | |
| *Atylus swammerdami* (Milne Edwards, 1830) | + | + | + | + | | + | + | + |
| *Atylus vedlomensis* (Bate and Westwood, 1862) | + | + | + | + | + | + | + | + |
| **Bathyporeidae** | | | | | | | | |
| *Bathyporeia elegans* Watkin, 1938 | + | + | + | + | | + | + | + |
| *Bathyporeia gracilis* Sars, 1981 | + | | * | * | | | * | + |
| *Bathyporeia guilliamsoniana* (Bate, 1857) | + | + | + | + | | + | + | + |
| *Bathyporeia nana* Toulmond, 1966 | + | | * | | | + | | |
| *Bathyporeia pelagica* (Bate, 1856) | + | + | + | + | | + | + | + |
| *Bathyporeia pilosa* Lindström, 1855 | + | * | + | * | | + | + | |
| *Bathyporeia sarsi* Watkin, 1938 | + | + | + | * | | + | + | |
| *Bathyporeia tenuipes* Meinert, 1877 | + | + | * | * | | | + | + |
| **Calliopiidae** | | | | | | | | |
| *Apherusa bispinosa* (Bate, 1857) | + | + | + | + | + | + | + | + |
| *Apherusa cirrus* (Bate, 1862) | + | + | + | * | | + | + | |
| *Apherusa clevei* G.O. Sars, 1904 | + | + | + | + | + | | + | |
| *Apherusa henneguyi* Chevreux & Fage, 1925 | | + | * | | | | | |
| *Apherusa jurinei* Milne-Edwards, 1830 | + | + | + | + | | + | + | + |
| *Apherusa ovalipes* Norman & Scott, 1906 | + | + | + | + | + | + | + | |
| *Calliopius laeviusculus* (Krøyer, 1838) | + | + | * | + | | + | + | |
| **Caprellidae** | | | | | | | | |
| *Caprella acanthifera* Leach, 1814 | + | + | + | + | | | | |
| *Caprella equilibra* Say, 1818 | | + | + | * | | | | |
| *Caprella erethizon* Mayer, 1909 | + | + | + | + | | | + | |
| *Caprella fretensis* Stebbing, 1878 | + | + | + | + | | | | |
| *Caprella linearis* (Linneus, 1767) | + | + | + | + | | | + | |
| *Caprella mutica* Schurin, 1935—NIS | | ** | ** | ** | | ** | | ** |
| *Caprella penantis* (Leach, 1814) | + | + | + | + | | | + | |
| *Caprella septentrionalis* Krøyer, 1838 | | | ** | | | | | |
| *Caprella tuberculata* Guérin, 1836/Bate & Westwood, 1866 | + | + | + | + | | | + | |
| *Pariambius typicus* Krøyer, 1845 | + | + | + | + | | | + | + |
| *Parvipalpus capillaceus* Chevreux, 1888 | | | ** | | | | | |
| *Phtisica marina* Slabber, 1769 | + | + | + | + | + | | + | + |
| *Pseudoprotella phasma* (Montagu, 1804) | + | + | + | + | + | | + | |
| *Pseudolirius kroyeri* (Haller, 1879) | | + | | + | | | | |
| **Cheirocratidae** | | | | | | | | |
| *Cheirocratus assimilis* (Lilljeborg, 1852) | + | + | + | + | + | | + | + |
| *Cheirocratus intermedius* G.O. Sars, 1894 | + | + | + | * | + | | + | + |
| *Cheirocratus sundevallii* (Rathke, 1843) | + | + | + | + | | + | + | + |
| **Cheluridae** | | | | | | | | |
| *Chelura terebrans* Philippi, 1839 | + | + | + | | | + | + | |
| **Colomastigidae** | | | | | | | | |
| *Colomastix pusilla* Grübe, 1861 | + | + | + | + | + | + | + | + |
| **Corophiidae** | | | | | | | | |
| *Apocorophium acutum* Chevreux, 1908 | + | + | + | | | + | | |
| *Chelicorophium curvispinum* (G.O. Sars, 1895)—NIS | | | | ** | | | | |

**Table 2.** *Cont.*

| | FWEC | EWEC | NBG | BS | CEC | WIGHT | FEEC | EEEC |
|---|---|---|---|---|---|---|---|---|
| *Corophium arenarium* Craword, 1937 | + | | * | * | | | + | |
| *Corophium volutator* (Pallas, 1766) | + | + | + | + | | + | + | |
| *Crassicorophium bonellii* (Milne-Edwards, 1830) | + | + | + | * | | | * | + |
| *Crassicorophium crassicorne* (Bruzelius, 1859) | + | + | + | * | + | | + | + |
| *Leptocheirus bispinosus* Norman, 1908 | + | | + | | | | | |
| *Leptocheirus hirsutimanus* (Spence Bate, 1862) | + | + | + | + | + | | + | + |
| *Leptocheirus pectinatus* (Norman, 1869) | + | + | + | + | | | * | + |
| *Leptocheirus pilosus* (Zaddach, 1844) | | + | + | + | | | | |
| *Leptocheirus tricristatus* (Chevreux, 1887) | + | + | + | | | | | |
| *Medicorophium runicorne* (Della Valle, 1893) | | | ** | | | | | |
| *Monocorophium acherusicum* (Costa, 1853)—NIS | + | + | + | + | | + | | |
| *Monocorophium insidiosum* (Crawford, 1937) | | + | + | * | | | + | |
| *Monocorophium sextonae* (Crawford, 1937)—NIS | + | + | + | + | + | + | + | + |
| **Cressidae** | | | | | | | | |
| *Cressa dubia* (Bate, 1857) | + | + | + | + | + | + | + | |
| **Cyproideidae** | | | | | | | | |
| *Peltocoxa brevirostris* (Scott and Scott, 1893) | + | + | + | | | | | |
| *Peltocoxa damnoniensis* (Stebbing, 1885) | + | + | + | | + | + | | |
| **Dexaminidae** | | | | | | | | |
| *Dexamine spinosa* (Montagu, 1813) | + | + | + | + | | + | + | |
| *Dexamine thea* Boeck, 1861 | + | + | + | + | | + | + | |
| *Guernea (Guernea) coalita* (Norman, 1868) | + | + | + | + | | | + | |
| *Tritaeta gibbosa* (Bate, 1862) | + | + | + | + | + | + | + | |
| **Dulichiidae** | | | | | | | | |
| *Dyopedos monacanthus* (Metzger, 1875) | + | | + | * | + | | | |
| *Dyopedos porrectus* (Bate, 1857) | + | | + | + | | | + | |
| **Epimeriidae** | | | | | | | | |
| *Epimeria cornigera* (Fabricius, 1779) | | + | + | | | | | |
| **Eriopisidae** | | | | | | | | |
| *Eriopisella pusilla* Chevreux, 1920 | + | | * | | | | | |
| **Eusiridae** | | | | | | | | |
| *Eusirus longipes* Boeck, 1861 | + | + | + | * | + | | * | + |
| **Gammarellidae** | | | | | | | | |
| *Gammarellus angulosus* (Rathke, 1843) | + | + | + | | + | | + | |
| *Gammarellus homari* (Fabricius, 1779) | | + | + | + | + | | + | |
| **Gammaridae** | | | | | | | | |
| *Dikerogammarus villosus* (Sowinsky, 1894)—NIS | | | | ** | | | | |
| *Echinogammarus berilloni* (Catta, 1878) | | | | | | | ** | |
| *Echinogammarus marinus* (Leach, 1815) | + | + | + | + | | + | + | |
| *Echinogammarus obtusatus* (Dahl, 1938) | + | + | + | | | + | + | |
| *Echinogammarus pirloti* (Sexton & Spooner, 1940) | | + | + | * | | + | | |
| *Echinogammarus stoerensis* (Reid, 1938) | | + | + | | | | + | |
| *Gammarus chevreuxi* Sexton, 1913 | + | + | * | | | | | |
| *Gammarus crinicornis* Stock, 1966 | + | + | + | * | | | + | |
| *Gammarus duebeni* (Lijeborg, 1852) | + | + | + | + | | + | + | + |
| *Gammarus finmarchicus* (Dahl, 1938) | | + | + | | | + | | |
| *Gammarus insensibilis* Stock, 1966 | | | * | * | | + | | |
| *Gammarus locusta* (Linnaeus, 1758) | + | + | + | + | | + | + | |
| *Gammarus oceanicus* (Segerstrale, 1947) | + | + | * | | | | | |
| *Gammarus salinus* (Spooner, 1947) | + | | * | * | | + | + | |
| *Gammarus zaddachi* (Sexton, 1912) | + | + | * | * | | + | + | + |
| *Homoeogammarus planicrurus* (Reid, 1940) | | + | * | | | + | | |
| **Haustoriidae** | | | | | | | | |
| *Haustorius arenarius* (Slabber, 1769) | + | + | + | * | | + | + | + |
| **Hyalidae** | | | | | | | | |
| *Apohyale prevostii* (Milne-Edwards, 1830) | + | + | + | + | | + | + | |
| *Hyale perieri* (Lucas, 1849) | + | + | + | | | | | |
| *Hyale pontica* Rathke, 1847 | + | + | + | | | + | | |

**Table 2.** *Cont.*

| | FWEC | EWEC | NBG | BS | CEC | WIGHT | FEEC | EEEC |
|---|---|---|---|---|---|---|---|---|
| *Hyale stebbingi* Chevreux, 1888 | + | | | | | | | |
| *Ptilohyale littoralis* (Stimpson, 1853)—NIS | | | | | | | ** | |
| **Hyperiidae** | | | | | | | | |
| *Hyperia galba* (Montagu, 1813) | + | + | + | + | | | + | |
| *Hyperia medusarum* (Müller, 1776) | | + | | | | | | |
| *Themisto abyssorum* (Boeck, 1871) | + | | | | | | | |
| *Themisto gaudichaudi* Guérin, 1825 | | + | | | | | | |
| **Iphimediidae** | | | | | | | | |
| *Iphimedia eblanae* Bate, 1857 | + | + | + | + | + | | + | + |
| *Iphimedia minuta* (G.O. Sars, 1882) | + | + | + | + | + | | + | + |
| *Iphimedia nexa* Myers & McGrath, in Myers, McGrath & Costello, 1987 | + | | + | + | | | + | |
| *Iphimedia obesa* (Rathke, 1843) | + | + | + | + | + | + | + | |
| *Iphimedia perplexa* Myers & McGrath, in Myers, McGrath & Costello, 1987 | + | + | + | * | | | + | |
| *Iphimedia spatula* Myers & McGrath, in Myers, McGrath & Costello, 1987 | + | + | + | + | | | + | |
| **Isaeidae** | | | | | | | | |
| *Isaea elmhirsti* Patience, 1909 | + | + | + | | | | | |
| *Isaea montagui* (Milne-Edwards, 1830) | + | + | + | | | + | + | |
| **Ischyroceridae** | | | | | | | | |
| *Centraloecetes kroyeranus* (Spence Bate, 1857) | + | + | + | + | | + | + | + |
| *Centraloecetes striatus* (Myers & McGrath, 1979) | + | | * | | | | | |
| *Cerapus crassicornis* (Bate, 1856) | | | + | | | | | |
| *Ericthonius difformis* (Milne-Edwards, 1830) | + | + | + | | | | | |
| *Ericthonius punctatus* (Bate, 1857) | + | + | + | + | + | + | + | + |
| *Ischyrocerus anguipes* Krøyer, 1838 | + | + | + | + | + | | + | |
| *Jassa falcata* (Montagu, 1808) | + | + | + | + | | + | + | |
| *Jassa herdmani* (Walker, 1893) | | | * | | | | + | |
| *Jassa ocia* (Bate, 1862) | + | | + | | | | | |
| *Jassa marmorata* Holmes, 1905 | + | | | * | | + | + | |
| *Jassa pusilla* (Sars, 1894) | + | | | + | | | + | |
| *Jassa slatteryi* Conlan, 1990 | | | ** | | | | | |
| *Microjassa cumbrensis* (Stebbing & Robertson, 1891) | | + | + | + | | | + | |
| *Parajassa pelagica* (Leach, 1814) | + | + | + | | | | | |
| **Lafystiidae** | | | | | | | | |
| *Lafystius sturionis* Krøyer, 1842 | | + | + | | | | | |
| **Leucothoidae** | | | | | | | | |
| *Leucothoe incisa* (Robertson, 1892) | + | + | + | + | + | | + | + |
| *Leucothoe lilljeborgi* Boeck, 1861 | + | + | + | + | | | + | |
| *Leucothoe procera* Bate, 1857 | + | + | + | + | | | + | + |
| *Leucothoe spinicarpa* (Abilggaard, 1789) | + | + | + | + | + | + | + | + |
| **Liljeborgiidae** | | | | | | | | |
| *Idunella dentipalma* (Dauvin & Gentil, 1983) | + | | | | | | | |
| *Idunella picta* (Norman, 1889) | + | + | + | + | | | + | |
| *Idunella spinifera* (Dauvin & Gentil, 1983) | + | | | | | | | |
| *Liljeborgia kinahani* (Bate, 1862) | + | | + | | | + | | |
| *Liljeborgia pallida* (Bate, 1857) | + | + | + | + | + | | + | |
| *Sextonia longirostris* Chevrreux, 1920 | + | | + | | | | | |
| **Lysianassidae** | | | | | | | | |
| *Lysianassa ceratina* (Walker, 1889) | + | + | + | + | | + | + | |
| *Lysianassa insperata* Lincoln, 1979 | + | | + | * | | | | |
| *Lysianassa plumosa* Boeck, 1871 | + | + | + | | + | | | |
| *Nannonyx goesii* (Boeck, 1871) | + | + | + | | | | | |
| *Nannonyx spinimanus* Walker, 1895 | | | | + | | | | |
| *Socarnes erythrophthalmus* Robertson, 1892 | + | + | + | + | + | | + | + |
| *Socarnes filicornis* (Heller, 1866) | + | + | + | | | | | |

**Table 2.** *Cont.*

| | FWEC | EWEC | NBG | BS | CEC | WIGHT | FEEC | EEEC |
|---|---|---|---|---|---|---|---|---|
| **Maeridae** | | | | | | | | |
| *Animoceradocus semiserratus* (Bate, 1862) | + | + | + | + | + | | * | + |
| *Elasmopus rapax* Costa, 1853 | + | + | + | * | | | | |
| *Elasmopus thalyae* Gouillieux & Sorbe, 2015 | | | ** | | | | | |
| *Maera grossimana* (Montagu, 1808) | + | + | + | + | + | + | + | |
| *Maera loveni* (Bruzelius, 1859) | | ** | | | | | | |
| *Maerella tenuimana* (Bate, 1862) | + | + | + | + | + | | + | |
| *Othomaera othonis* (H. Milne Edwards, 1830) | + | + | + | + | + | + | + | + |
| *Quadrimaera inaequipes* (A. Costa in Hope, 1851) | + | | | | | | | |
| **Megaluropidae** | | | | | | | | |
| *Megaluropus agilis* Hoeck, 1889 | + | + | + | + | | + | + | |
| **Melitidae** | | | | | | | | |
| *Abludomelita gladiosa* (Bate, 1862) | + | + | + | + | + | | + | + |
| *Abludomelita obtusata* (Montagu, 1813) | + | + | + | + | | | + | + |
| *Allomelita pellucida* (Sars, 1882) | | + | + | + | | | * | + |
| *Melita hergensis* Reid, 1839 | + | + | + | | | + | | |
| *Melita palmata* (Montagu, 1804) | + | + | + | + | | + | + | |
| **Melphidippidae** | | | | | | | | |
| *Melphidippella macra* (Norman, 1869) | + | + | + | + | + | | + | |
| **Microprotopidae** | | | | | | | | |
| *Microprotopus longimanus* Chevreux, 1887 | + | | + | + | | | + | |
| *Microprotopus maculatus* Norman, 1867 | + | + | + | + | | + | + | |
| **Nuuanuidae** | | | | | | | | |
| *Gammarella fucicola* (Leach, 1814) | + | + | + | * | | + | | |
| **Odiidae** | | | | | | | | |
| *Odius carinatus* (Spence Bate, 1862) | | | + | | | | | |
| **Oedicerotidae** | | | | | | | | |
| *Deflexilodes subnudus* (Norman, 1889) | + | + | + | | + | | | |
| *Deflexilodes tuberculatus* (Boeck, 1871) | + | | | | | | | |
| *Kroyera carinata* Spence Bate, 1857 | + | + | + | + | | | | |
| *Perioculodes longimanus* (Bate & Westwood, 1868) | + | + | + | + | + | + | + | |
| *Pontocrates altamarinus* (Bate & Westwood, 1862) | + | + | + | + | | + | + | + |
| *Pontocrates arenarius* (Bate, 1858) | + | + | + | + | + | + | + | + |
| *Synchelidium haplocheles* (Grübe, 1864) | + | + | + | | | + | | |
| *Synchelidium maculatum* Stebbing, 1906 | + | + | + | + | + | | + | + |
| *Westwoodilla caecula* (Spence Bate, 1857) | | + | | | | | | |
| **Opisidae** | | | | | | | | |
| *Normanion chevreuxi* Diviacco & Vader, 1988 | + | | | | | | | |
| **Phliantidae** | | | | | | | | |
| *Pereionotus testudo* (Montagu, 1808) | | + | + | | | | | |
| **Photidae** | | | | | | | | |
| *Gammaropsis lobata* (Chevreux, 1920) | | + | | | | | | |
| *Gammaropsis maculata* (Johnston, 1828) | + | + | + | + | + | + | + | + |
| *Gammaropsis nitida* (Stimpson, 1853) | + | | | * | | + | + | + |
| *Gammaropsis palmata* (Stebbing & Robertson, 1891) | + | + | + | | | | | |
| *Gammaropsis sophiae* (Boeck, 1861) | | + | + | | | | | |
| *Megamphopus cornutus* Norman, 1869 | + | + | + | + | + | + | + | |
| *Photis longicaudata* (Bate & Westwood, 1862) | + | + | + | + | | + | + | + |
| *Photis pollex* Walker, 1895 | | ** | | | | | | |
| *Photis reinhardi* Krøyer, 1842 | | | | | | | + | |
| **Phoxocephalidae** | | | | | | | | |
| *Harpinia antennaria* Meinert, 1890 | | + | + | + | | + | + | |
| *Harpinia crenulata* (Boeck, 1871) | + | + | * | + | | | | |
| *Harpinia pectinata* Sars, 1891 | + | | + | | | | | |
| *Metaphoxus fultoni* (Scott, 1890) | + | + | + | + | + | | + | + |
| *Metaphoxus simplex* (Bate, 1857) | | + | + | * | | | | |

**Table 2.** *Cont.*

| | FWEC | EWEC | NBG | BS | CEC | WIGHT | FEEC | EEEC |
|---|---|---|---|---|---|---|---|---|
| **Pleustidae** | | | | | | | | |
| *Parapleustes bicuspis* (Krøyer, 1838) | | | + | + | + | | + | |
| *Stenopleustes latipes* (M. Sars, 1858) | | | + | | | | | |
| *Stenopleustes nodifera* (G.O. Sars, 1883) | + | + | + | | + | | + | |
| **Podoceridae** | | | | | | | | |
| *Podocerus variegatus* Leach, 1814 | + | + | + | + | | + | | |
| **Scopelocheiridae** | | | | | | | | |
| *Scopelocheirus hopei* (Costa in Hope, 1851) | + | + | + | + | | | | |
| **Stegocephalidae** | | | | | | | | |
| *Stegocephaloides christianiensis* Boeck, 1871 | | | * | | | | | |
| **Stenothoidae** | | | | | | | | |
| *Metopa alderi* (Spence Bate, 1857) | | | | * | | + | + | |
| *Metopa borealis* GO Sars, 1892 | | | + | | | | + | + |
| *Metopa bruzelii* (Goes, 1866) | ** | | | | | | | |
| *Metopa pusilla* GO Sars, 1892 | | | + | | | | + | + |
| *Metopa tenuimana* GO Sars, 1892 | | | | + | | | | |
| *Metopa rubrovittata* G.O. Sars, 1883 | + | | + | + | | | + | |
| *Parametopa kervillei* Chevreux, 1901 | + | + | + | + | + | | + | |
| *Stenothoe ascidiae* (Pirlot, 1933) | + | | | | | | | |
| *Stenothoe eduardi* Krapp-Schickel, 1975 | + | + | + | | | | | |
| *Stenothoe marina* (Spence Bate, 1857) | + | + | + | + | + | + | + | + |
| *Stenothoe monoculoides* (Montagu, 1815) | + | + | + | + | | + | + | + |
| *Stenothoe tergestina* (Nebeski, 1881) | + | + | + | | | | | |
| *Stenothoe valida* Dana, 1852 | | + | * | | | | * | |
| **Synopiidae** | | | | | | | | |
| *Austrosyrrhoe fimbriatus* (Stebbing & Robertson, 1891) | + | | | | | | | |
| **Talitridae** | | | | | | | | |
| *Britorchestia brito* (Stebbing, 1891) | | | * | | | | + | |
| *Cryptorchestia cavimana* (Heller, 1865)—NIS | | | | ** | | | | |
| *Deshayesorchestia deshayesii* (Audouin, 1826) | + | + | + | * | | | | |
| *Orchestia gammarellus* (Pallas, 1776) | + | + | + | + | | + | + | |
| *Orchestia mediterranea* Costa, 1853 | + | + | + | + | | + | + | |
| *Platorchestia platensis* (Krøyer, 1845)—NIS | | | | ** | | | | |
| *Talitrus saltator* (Montagu, 1808) | + | + | + | + | | + | + | |
| **Tryphosidae** | | | | | | | | |
| *Hippomedon denticulatus* (Bate, 1857) | + | + | + | + | + | | + | + |
| *Hippomedon oculatus* Chevreux & Fage, 1925 | | | + | | | | | |
| *Lepidepecreum longicorne* (Bate & Westwood, 1862) | + | + | + | + | | | + | |
| *Orchomene humilis* (Costa, 1853) | + | + | + | + | + | + | + | |
| *Orchomene similis* (Chevreux, 1912) | + | | + | | | | | |
| *Orchomenella commensalis* Chevreux & Fage, 1925 | | | | + | | | | |
| *Tryposa crenata* (Chevreux & Fage, 1925) | + | | | | | | | |
| *Tryphosa nana* (Kröyer, 1846) | + | + | + | + | + | + | + | + |
| *Tryphosella horingi* (Boeck, 1871) | + | + | * | | | | + | |
| *Tryphosella minima* (Chevreux, 1911) | + | | | | | | | |
| *Tryphosella nanoides* (Lilljeborg, 1865) | | + | | | + | | | |
| *Tryphosella sarsi* Bonnier, 1893 | + | + | + | + | + | | + | |
| *Tryphosites longipes* (Spence Bate & Westwood, 1861) | + | | + | * | | | | |
| **Unciolidae** | | | | | | | | |
| *Uncinotarsus pellucidus* L'Hardy and Truchot, 1964 | + | | | | | | | |
| *Unciola crenatipalma* (Spence Bate, 1862) | + | + | + | + | + | + | + | + |
| **Uristidae** | | | | | | | | |
| *Ichnopus spinicornis* Boeck, 1861 | + | | | | | | | |
| *Menigrates obtusifrons* (Boeck, 1861) | | | + | | | | | |
| *Tmetonyx cicada* (Fabricius, 1780) | | + | + | + | | | + | |
| *Tmetonyx similis* (G.O. Sars, 1891) | + | | + | | + | | + | |

**Table 2.** *Cont.*

|  | FWEC | EWEC | NBG | BS | CEC | WIGHT | FEEC | EEEC |
|---|---|---|---|---|---|---|---|---|
| **Urothoidae** | | | | | | | | |
| *Urothoe brevicornis* Bate, 1862 | + | + | + | + |  | + | + | + |
| *Urothoe elegans* (Bate, 1857) | + | + | + | + | + |  | + | + |
| *Urothoe grimaldii* Chevreux, 1895 | + |  | + | + |  |  |  |  |
| *Urothoe marina* (Bate, 1857) | + | + | + | + | + | + | + | + |
| *Urothoe poseidonis* Reibish, 1905 | + | + | + | * |  | + | * | + |
| *Urothoe pulchella* (Costa, 1853) | + |  | + | * |  |  | * | + |

Two species, *Aora spinicornis* (Afonso, 1976) and *Metopa bruzelii* (Goes, 1866), are newly recorded for the French side of the Western EC from the *Laminaria hyperborea* kelp of the Roscoff zone [16].

Six species are added for the English side of the Western EC: *Caprella mutica* Schurin, 1935, from Plymouth harbour [17]; this species was also present in the Normano-Breton Gulf, the Bay of Seine, and along the English coast, Wight, and the eastern EC, confirming that *C. mutica* is a well-established species in the EC [12,17,47,84]. The study of [18] has led to the addition of five new species to the English side of the Western EC: three Amphilochidae: *Amphilochoides boeckii* G.O. Sars, 1895, *Amphilochoides serratipes* (Norman, 1869), and *Paramphilochoides intermedius* (Scott, 1896); one Maeridae, *Maera loveni* (Bruzelius, 1859); and one Photidae, *Photis pollex* Walker, 1895. All these species are recorded around the coast of the British Isles [3].

Ten species are newly reported here for the NBG zone: *Ampelisca toulemonti* in the Normano-Breton Gulf [38] and in the North Cotentin [11]; *Aoroides longimerus* Ren & Zheng, 1996, and *Aoroides semicurvatus* Ariyama, 2004, *Caprella mutica* Schurin, 1935, in Granville harbour, and *A. semicurvatus* and *C. mutica* in Cherbourg harbour [9]; the caprellidae *Caprella septentrionalis* Krøyer, 1838, in the Chausey Islands and *Parvipalpus capillaceus* Chevreux, 1888, in the north-western Cotentin [14]; *Jassa slatteryi* in Dielette harbour (unpublished data); *Medicorophium runicorne* (Della Valle, 1893) in the Rade de Cherbourg (Sébastien Aubin, Muséum National d'Histoire Naturelle, Dinard, personal communication); and *Elasmopus thalyae* Gouillieux & Sorbe, 2015, from the Raz Blanchard in the western part of the English Channel to the North of the Normano-Breton Gulf in a zone with the strongest tidal currents in Europe [19].

Ten species are newly reported in the Bay of Seine zone: *Ampithoe valida* S.I. Smith, 1873, in *Sargassum muticum* macroalgae at Saint-Vaast La-Hougue [13]; *Aoroides longimerus* Ren & Zheng, 1996, and *Aoroides semicurvatus* Ariyama, 2004, in the harbour of Saint-Vaast-La-Hougue and Le Havre [9]; *Caprella mutica* Schurin, 1935, in the harbour of Le Havre; *Grandidierella japonica* Stephensen, 1938, along the coast of Calvados and near the Isle of Wight [12,23]; *Chelicorophium curvispinum* (G.O. Sars, 1895) in the lower part of the Seine estuary [85]; and *Dikerogammarus villosus* (Sowinsky, 1894), *Cryptorchestia cavimana* (Heller, 1865), and *Platorchestia platensis* (Krøyer, 1845) in the Seine estuary or the harbour basin of Le Havre [12].

For the Wight zone, only two NIS have been recorded recently: *Caprella mutica* Schurin, 1935, and *Grandidierella japonica* Stephensen, 1938 [17,23].

For the eastern part of the EC, three species were added to the checklist of [1]: two NIS *Ptilohyale littoralis* (Stimpson, 1853) [10,58] and *Caprella mutica* Schurin, 1935 [17], and *Echinogrammarus berilloni* (Catta, 1878), sampled in the upper part of the Le Tréport harbour (unpublished data).

The checklist of [1] includes 255 species of amphipods in the EC (i.e., 240 Gammaridea, four Hyperiidea, and eleven Caprellidea). Among the 255 species reported, *Anonyx* sp. (Kröyer, 1838) and *Lysianassa* sp. (H. Milne Edwards, 1830) are probably new species, but in the absence of new records or descriptions, they are not considered in the current checklist.

Furthermore, in the 1999 checklist, seven species were reported as doubtful in the EC (Table 3). Thus, only one species, *Odius carinatus* (Spence Bate, 1862), listed as doubtful in the 1999 checklist in the EC has been confirmed as present in the EC from the first checklist of EC Amphipoda [14].

**Table 3.** Species reported as doubtful in the English Channel in [1].

| Species | Zone | Record | Reason for Discounting |
|---|---|---|---|
| *Ampelisca gibba* Sars, 1883 | Guernsey | [86] | A deep-sea species |
| *Menigrates obtusifrons* (Boeck, 1861) | Guernsey | [86] | A boreal species |
| *Monoculodes tesselatus = Deflexilodes tesselatus* (Schneider, 1883) | Plymouth | [5] | A boreal species |
| *Proboloides gregaria* (G.O. Sars, 1883) | Bay of Seine | [2] | A boreal species |
| *Phoxocephalus holbolli* (Krøyer, 1842) | Plymouth | [5] | A boreal species |
| *Stenopleustes malmgreni* (Boeck, 1871) | Plymouth | [5] | = *Stenopleustes nodifera* (G.O. Sars, 1883). |

In fact, Ref. Dauvin (1999) [1] gives 232 as the number of confirmed species of Gammaridea in place of 240, and the total number of amphipod species as 247 in place of 255.

One species, *Gammarus insensibilis*, has been reported outside the five main zones inventoried by [1]; it has been recorded in the English Channel from the Portland zone (Wight) by Lincoln (1979). This species has been reported in the Normano-Breton Gulf by [14] and is also found in the Bay of Seine. Furthermore, two species, *Protohyale (Protohyale) grimaldii* (Chevreux, 1891) (= *Hyale grimaldii*) and *Jassa slatteryi* Conlan, 1990, show a large geographical distribution extended to the north and south of the EC. In 1999, both species were reported as probably present in the EC but were not included in the 1999 checklist (see Gouilleux, 2017, for distribution of *J. slatteryi* along the French coast of the north-eastern Atlantic). *J. slatteryi* was recognized as present in the Normano-Breton Gulf during a survey of the colonization of panels placed in Dielette harbour (unpublished data).

### 3.2. Non-Indigenous Species

Twelve NIS have been reported in the EC, including eight that were recently recorded: *Aoroides longimerus* Ren & Zheng, 1996, *Aoroides semicurvatus* Ariyama, 2004, *Ampithoe valida* S.I. Smith, 1873, *Caprella mutica* Schurin, 1935, *Chelicorophium curvispinum* (G.O. Sars, 1895), *Dikerogammarus villosus* (Sowinsky, 1894), *Grandidierella japonica* Stephensen, 1938, and *Ptilohyale littoralis* (Stimpson, 1853) [9,10,12,13,17,23,47,84,87–89]. Four other species have been known in the EC for several decades: *Cryptorchestia cavimana* (Heller, 1865) since 1959 in the Seine estuary, *Monocorophium acherusicum* (Costa, 1853), reported in 1965, *Monocorophium sextonae* (Crawford, 1937) reported in 1976 in the Bay of Seine, and *Platorchestia platensis* (Krøyer, 1845) reported in the same zone at the beginning of the 1990s in Normandy [89].

Four other NIS have been reported in neighbouring regions of the EC: *Aoroides curvipes* Ariyama, 2004, *Gammarus tigrinus* Sexton, 1939, *Melita nitida* Smith, 1873, and *Monocorophium uenoi* (Stephensen, 1932). Therefore, they could be recorded in the following years in the EC [3,43,55,74,90].

### 3.3. Pattern Distribution of Amphipod Fauna in the English Channel

The 269 species listed here belong to 57 families, with 21 of them containing only a single species and one family alone accounting for 16 species (Figure 2). Only eight families comprise more than 10 species (Figure 2). A total of 44 species (16%) are recorded in only one of the eight delimited zones, with 21 being found in two zones and 29 in three zones. Most of the species recorded in one, two, or three zones are found only in the three zones of the western basin of the EC. A total of 14 species are recorded in all eight zones of the EC, with 43 other species in seven out of the eight zones; most of these species are absent in the Central Part of the EC (15 species), the Wight zone (18 species), and on the English side of the eastern EC (10 species): i.e., in the three zones with the lowest number of reported species.

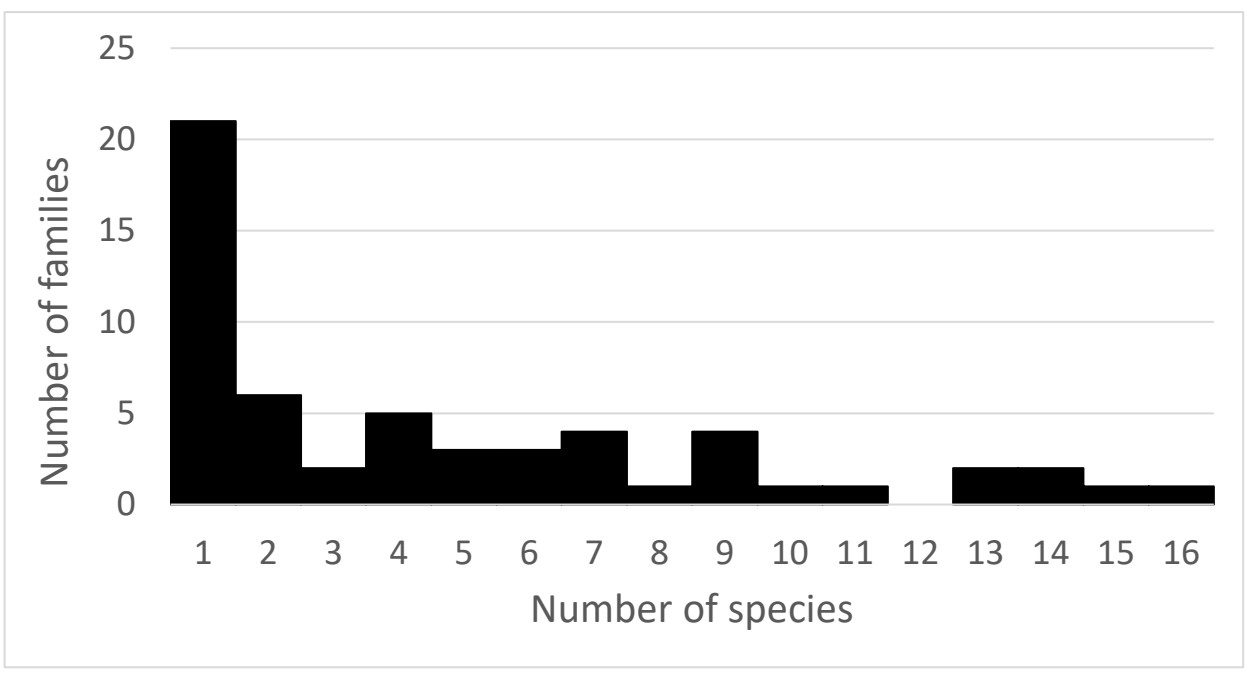

**Figure 2.** Distribution of the numbers of species belonging to the 57 amphipod families recorded in the English Channel.

In the Normano-Breton Gulf, the total number of recorded species reaches 224, while 201 are recorded on the French side of the western EC, 194 on the English side, 172 in the Bay of Seine, 149 on the French side of the eastern EC, 97 in the Wight zone, 64 on the English side of the eastern EC, and finally 61 in the central part of the EC. In comparison with the 1999 checklist, the numbers of species have markedly increased in the Normano-Breton Gulf (addition of 30 species) due to the publication of the Atlas of invertebrates of this zone by [14]. This increase is also observed in harbours of the North-Cotentin and the Bay of Seine (addition of 42 species), where additional studies have been carried out in relation to numerous surveys linked to anthropic activities. For the French side of the eastern EC, the number of additional species reaches 13, while the number of species in the western part of the EC remains similar between the 1999 and 2022 checklists.

When applied to the eight zones and the 269 species recorded in the EC, Hierarchical Cluster Analysis (HCA) shows that we can separate the Central EC and the English side of the Eastern EC from the six other zones at a level of similarity of 50% (Figure 3). At a level of 40%, Wight is separated from the five other zones, while, at a level of 20%, we can distinguish the eastern basin of the EC (Bay of Seine and French side of the eastern EC) from the three zones of the western basin of the EC.

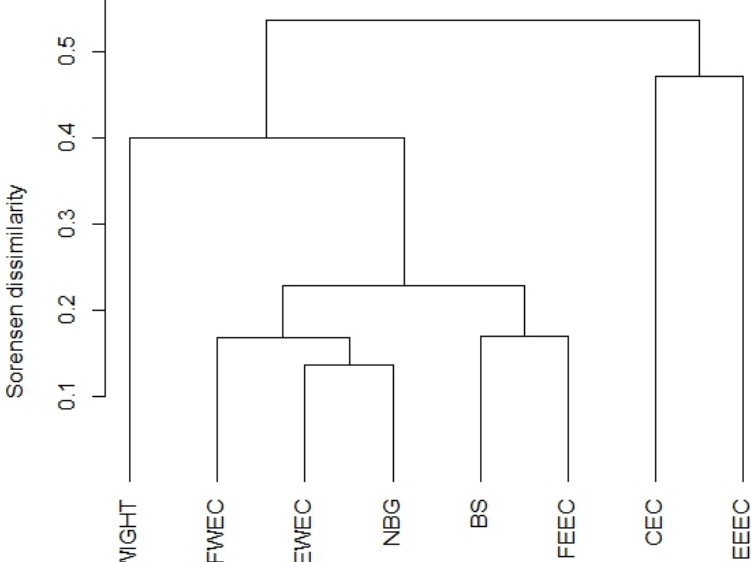

**Figure 3.** Results of cluster analysis (Sørensen similarity) on the species presence and absence (269 species) in the eight zones of the English Channel: FWEC: French side of Western English Channel; EWEC; English side of Western English Channel; NBG: Normano-Breton Gulf; BS: Bay of Seine; CEC: Central English Channel; Wight; FEEC: French side of Eastern English Channel; EEEC: English side of Eastern English Channel.

## 4. Discussion

The number of amphipod species collected in the English Channel was 269. This census significantly increases the historical number of known amphipods of the English Channel; 162 species were listed for the English Channel in the Fauna of the Amphipods of France [2] out of a total of 320 species known at the beginning of the 20th century for the whole of France including fresh waters. The Fauna of the Amphipods of the British Isles [3] listed 186 species of Gammaridea in the English Channel out of a total of 271 species for the whole of the British Isles, including 135 for the Normano-Breton Gulf and 161 for the Plymouth region on the English side of the western EC, slightly less than the number given in the inventory of Plymouth amphipods: 179 [5]. Reference [4] lists 145 species for Roscoff, while Bertrand and Retière counted 138 species for the Normano-Breton Gulf, and, finally, Glaçon drew up a checklist of 92 amphipods for the fauna of Wimereux (see [1]).

Among the 12 non-indigenous species recorded in the EC, eight had been reported since 1999 and had been mainly discovered in marinas, harbours, and shallow waters during samplings devoted to the knowledge of the fauna living in hard-bottom habitats. The pathways of their recent introductions were attributed to oyster transfers and ship traffics on both sides of the English Channel [12,17,88].

The three zones of the western basin of the EC account for more species than the eastern basin of the EC (Table 2). Out of a total of 269 species, 180 (67%) have been recorded in both basins of the EC and therefore display a wide geographical distribution, while 78 species have been reported only in the western basin and 11 species solely in the eastern basin. Even if fewer surveys have been carried out in the eastern compared to the western English Channel, there appears to be a west-to-east impoverishment of the amphipod fauna of the EC—as observed for other zoological groups such as polychaetes and mollusc [91]—except in the Bay of Seine, where there has been a great increase in the number of species over the last two decades, probably due to the increase in collection in this zone (130 in 1999 as against 173 in 2022). This impoverishment appears to be in response to a climatic gradient from Atlantic Ocean waters, with low annual thermohaline variability in the west to waters of much more continental character and high annual

thermohaline variability in the east [92]. Moreover, it is clear that the poorest zones in terms of species diversity are those with the fewest studies, such as in the central part of the EC. However, many physical, biological, and physiological factors such as temperature, current and sediment patterns, depth, and competition and interactions between species could also influenced the presence of amphipods in the eight zones of the English Channel.

Unless there is a significant climate change with a high increase of sea-water temperature allowing the spread of warm-amphipod species into the EC, the number of amphipod species present in the English Channel is expected to remain in the same order of magnitude as estimated for the British Isles as a whole (about 290–300 species). Nevertheless, in the future, we expect to see an increase in the numbers of species reported in the three zones poorest in amphipods because new studies will cover the central part of the EC as well as two zones on the English side of the eastern EC. It is also likely that new records could be concerned with species recently described or reported in neighbouring zones, mainly along the French coast of the Bay of Biscay [93–95] and around the coasts of Ireland (see [29,33,34,37,41,42,45,53,63,64,66,68,69,71,75,77–79,81,82]), i.e., approximately 20 species (Table 2). Particular attention to amphipod identification by engineers from consulting offices and researchers from university laboratories on amphipod collections from new studies should lead to the discovery of these potential species.

**Funding:** This research received no external funding.

**Institutional Review Board Statement:** Not applicable.

**Data Availability Statement:** Not applicable.

**Acknowledgments:** My thanks to the Caen Normandy University for granting me Emeritus status and to the CNRS for their welcome to the 'Laboratoire Morphodynamique Continentale et Côtière'. This work is intended as a contribution to the knowledge of the amphipod fauna of the north-eastern Atlantic Ocean. I would like to express my gratitude to Eric L'Ebrellec for extracting information on amphipods from the MABES database (Macrobenthos de la Baie et Estuaire de Seine), Nathan Chauvel for the statistical analyses, and Michael Carpenter for post-editing the English style and grammar. I also thank the four reviewers and Sabrina Lo Brutto, editor of this special issue, for their very useful comments on the first versions of this paper.

**Conflicts of Interest:** The authors declare no conflict of interest.

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
