# Peer review of "An Update of Amphipoda Checklist for the English Channel"

_diversity, doi:10.3390/d14100783_

Round 1
Reviewer 1 Report
Congratulations to the authors for the work, I have not suggestion.
Author Response
Thanks to the reviewers
Reviewer 2 Report
Line 6. An up-to-date checklist for 2022…. Change to: An updated checklist of amphipods from the English Channel (La Manche). This revision brings up-to-date the inventory of Dauvin (1999) with recent data from Le Mao et al. (2019) and other studies focused on non-native fauna.
Line 7 “areas or zones”. It is not necessary to use both terms. use one or the other term throughout.
Line 11 ‘compared with’ change to ‘ than it is in’
Line 11 or the central part change to “or in the”
Line 23–24. More than twenty years ago, a checklist of amphipods recorded in the English channel (EC) was published by Dauvin (1999). It was based on data from Chevreux & Fage (1925)…..….
Line 40 ‘swimming’ change to ‘mobile’
Line 56 ‘as confident’ change to ‘with confidence’
Lines 81-82 spacing problem
Line 83. The families and species are ordered according to alphabetical names. change to: The families are ordered alphabetically, the genera are ordered alphabetically within the families and the species are ordered alphabetically within the genera.
Lines 103 – 144. I would suggest that these seven species be listed in a table as doubtful records:
|
Species |
Area |
Record |
Reasons for discounting |
|
Ampelisca gibba Sars |
Guernsey |
Walker & Hornell (1896) |
A deep-sea species |
etc.
Table 2 ‘nor recorded’ should be ‘not recorded’
Line 285. ‘The number of amphipod species collected…..can be estimated as 269”
Why estimated? Surely you know the number collected and it is 269?
Line 267. ‘historic’ should be ‘historicaL’
Line 309. ‘There has been a great increase in the number of species over the last two decades’ How can we be sure that this is not either due to increased collecting or to increased taxonomic knowledge?
Author Response
I have taken into account all his (her) precious comments to improve this paper on amphipods from the English Channel. I have added a table as suggested.
Thanks to this reviewer for very interesting suggestions.
Reviewer 3 Report
Please, see the attached file.

Author Response
Thanks to this reviewer for his (her) very fine analysis of my first version. He (she) has pointed out very interesting weakeness on my first version and is a very important review. I have taken into account all his (her) precious comments to improve this paper on amphipods from the English Channel.
Reviewer 4 Report
This is a much needed update of a vital source of amphipod knowledge. It is well thought out, any revisions I have suggested are just that - suggestions.
Thank you for the opportunity to review the paper.
Author Response
Thanks to the reviewers